# Hepatitis E Virus (HEV) in Makkah, Saudi Arabia: A Population-Based Seroprevalence Study

**DOI:** 10.3390/v15020484

**Published:** 2023-02-09

**Authors:** Mai M. El-Daly, Rajaa Al-Raddadi, Amany Alharbi, Abdulrahman E. Azhar, Amjed M. Khallaf, Ahmed M. Hassan, Osama M. Alwafi, Omaima I. Shabouni, Thamir A. Alandijany, Tian-Cheng Li, Sherif A. El-Kafrawy, Alimuddin Zumla, Esam I. Azhar

**Affiliations:** 1Special Infectious Agents Unit-BSL3, King Fahd Medical Research Center, King Abdulaziz University, Jeddah 21362, Saudi Arabia; 2Department of Medical Laboratory Sciences, Faculty of Applied Medical Sciences, King Abdulaziz University, Jeddah 21589, Saudi Arabia; 3Community Medicine Department, Faculty of Medicine, King Abdulaziz University, Jeddah 21589, Saudi Arabia; 4Student Research and Innovation Unit, Faculty of Medicine, King Abdulaziz University, Jeddah 21362, Saudi Arabia; 5Ministry of Health, Makkah 24226, Saudi Arabia; 6Ministry of Health, Jeddah 22246, Saudi Arabia; 7Department of Virology II, National Institute of Infectious Diseases, Gakuen 4-7-1, Musashi-murayama, Tokyo 208-0011, Japan; 8Department of Infection, Division of Infection and Immunity, Centre for Clinical Microbiology, University College London, London NW1 2PG, UK; 9NIHR Biomedical Research Centre, University College London Hospitals, London NW1 2PG, UK

**Keywords:** HEV, Makkah, Saudi Arabia, seroprevalence, risk factors

## Abstract

Background: The Hepatitis E virus (HEV) is a common cause of viral hepatitis worldwide. Little is known about the seroprevalence of HEV in the general population of Saudi Arabia. Methods: A community-based cross-sectional HEV seroprevalence study was conducted in Makkah, Saudi Arabia. Anti-HEV IgG antibodies were detected in sera using an in-house ELISA. The frequency of HEV sageerology and its correlation with demographic, and environmental factors were evaluated. Results: Enrollment consisted of 1329 individuals, ages ranged from 8 to 88 years, the mean age was 30.17 years, the median age was 28yrs, and the male: female ratio was 1.15. The overall seroprevalence was 23.8% (316/1329). Males had significantly higher seroprevalence than females (66.1 vs. 33.9%; *p* < 0.001). Seroprevalence had significant correlations with age, occupation, and lack of regular water supply and housing conditions. Conclusions: This is the first HEV community-based seroprevalence study from Saudi Arabia. Results show that the HEV is endemic in Makkah and affects all age groups and occupations. HEV affects more males than females and those living in crowded accommodations without a regular supply of water. Further studies are required across all regions of Saudi Arabia to determine the country’s seroprevalence of active or past infection using tests for HEV IgG, HEV IgM antibodies and/or HEV RNA and underlying determinants of transmission.

## 1. Introduction

The Hepatitis E virus (HEV) is a positive-sense single-stranded RNA virus that belongs to the *genus Orthohepevirus* in the *Hepeviridae* family [1]. The virus is assigned eight genotypes (HEV-1 to HEV-8) which belong to the species *Paslahepevirus balayani* [2]. The viral genome is about 7.2 kB in length and is organized into three open reading frames with at least eight genotypes (HEV-1 to HEV-8) [3]. 

HEV is a common infection of humans worldwide, and sporadic outbreaks of acute viral hepatitis continue to occur globally [1]. The infection is mostly asymptomatic and self-limited in the population, but a case fatality rate of up to 30% has been recorded in individuals who have preexisting liver disease, are malnourished, immunocompromised, or are in the third trimester of pregnancy [4,5]. The symptoms include jaundice, vomiting, nausea, abdominal pain, malaise, and anorexia, and they last typically for less than a month with recovery in nearly all individuals with an effective immune system [5]. The World Health Organization (WHO) estimates that 20 million new HEV cases occur annually, with 44,000 associated fatalities reported in 2015 [6]. HEV is mainly transmitted through the fecal-oral route by drinking contaminated water in endemic areas where HEV genotypes 1 and 2 are most prevalent [7], or by consuming contaminated animal products, particularly pork, in industrialized countries where genotypes 3 and 4 are most prevalent [8,9]. Animal reservoirs for HEV transmission were also reported in wild pigs, mongooses, deer, rabbits, [8,9] and dromedary camels in the Middle East [10]. Vertical transmission from infected mothers to their infants can occur [11,12]. Preventive measures include simple hygiene measures, improving sanitation, and cooking the consumed animal products [13]. Available vaccines are not licensed globally, and currently, there is no specific treatment [5,12]. 

Although several countries in The Middle East have recorded medium to high HEV seroprevalence rates, reports on HEV genotypes are rare. HEV genotype 7 was first reported in dromedary camels by Woo et al. [14] in the UAE from feces samples. Later in 2016, chronic zoonotic transmission with HEV genotype 7 was reported in a liver transplant patient who regularly consumed camel meat and milk in the UAE [10]. We recently reported the genetic diversity of HEV genotype 7 in imported and domestic dromedary camels in Saudi Arabia [15]. Our results showed that the sequences generated from African domestic dromedary camel samples clustered with Genbank sequences from Kenya, Somalia, and UAE, while sequences generated from domestic dromedary camels clustered with Genbank sequences from UAE, and both domestic and imported sequences clustered away from isolates reported from Pakistan. Reports from Tunisia and Israel showed the detection of HEV genotypes 1 and 3 in wastewater samples [16,17] and reports from Israel showed HEV genotype 1 in acute HEV cases [18]. In Saudi Arabia, there is scanty information on the prevalence of HEV in the general population [19]. Reports utilizing small sample sizes and targeting selected populations such as blood donors and hemodialysis patients [20,21,22,23,24,25] indicate that HEV is endemic in Saudi Arabia. Makkah is the holiest city in the Muslim World, where millions of Muslims from around the world gather every year to perform the Hajj pilgrimage rituals at a certain time of the year and millions more to perform Umrah all year round. The city has an area of about 1200 km^2^ and a population of about 1.6 million inhabitants. In this study, we performed a community-based cross-sectional study to determine the seroprevalence and evaluate potential risk factors of hepatitis E virus infection in residents of Makkah, Saudi Arabia. 

## 2. Materials and Methods

### 2.1. Study Design and Site

This cross-sectional community-based household seroprevalence study was conducted in the city of Makkah, Saudi Arabia. 

#### 2.1.1. Ethical Considerations

The study protocol was approved by the Research and Clinical Studies Department, Directorate of Health Affairs, Jeddah Ministry of Health ethical review board (approval number A00284). The study objectives and protocols were explained to the participants or their legal representatives. Written consent was obtained prior to sample collection.

#### 2.1.2. Population and Sampling

The study included residents of Makkah city and a multi-stage stratified cluster sampling was used to include the participating households; each household comprised six individuals. In the first stage, Makkah city was divided into five clusters based on sociodemographic characteristics using a global positioning system. We tried to include subjects from various sociodemographic backgrounds—males/females, different age groups, living in various forms of housing, pursuing a variety of occupations. In stage two, clusters were sampled by random selection of one point out of each cluster. In stage three, all households located at the selected point were visited, and all occupants were invited to participate [26]. Each participant was asked to donate 5 mL of venous blood in a plain tube upon their approval. Blood specimens were allowed to clot at room temperature (20–25 °C) and centrifuged. The sera were separated and stored at –20 °C till testing. Subjects were recruited from various districts of Makkah—the city center, the east, the west, the north, and the south of the city.

A semi-structured questionnaire was used to collect sociodemographic data such as age, gender, housing type, etc., and environmental risk factors such as connection to the sewage network, and availability of clean water supply. The questionnaire was administered by one of the investigators.

### 2.2. Serological Assays

HEV IgG antibodies were detected in the collected samples using an in-house enzyme-linked immunoassay (ELISA), in accordance with Li et al. and El-Kafrawy et al. [27,28,29] with modifications. The VLPs (DcHEV-LPs) used for the detection of anti-HEV IgG antibodies expressed the partial ORF2 of DcHEV (HEV-7) by a recombinant baculoviruses expression system. We confirmed that the human IgG anti- HEV-1, HEV-3, HEV-4, and HEV IgG reacted to DcHEV-LPs and to all homologous HEV-LPs with similar titers [30]. The in-house ELISA uses HEV-LPs as an antigen and had previously shown a higher sensitivity for HEV IgM, IgA, and IgG than the commercial kits [31]. The 96-well ELISA plates were coated with 100 µL of purified HEV virus-like particles and incubated at 4 °C overnight. The plates were washed three times with PBS containing 0.05% tween-20 (PBS-T), blocked with 200 µL of 5% skim milk in PBS-T, and incubated for 1 h at 37 °C. One hundred µL of 1:200 diluted samples were added to each well and incubated for 1 h at 37 °C. Goat peroxidase-labeled anti-human IgG antibodies (100 µL) in a 1:16,000 dilution were added to the wells and incubated for 1 h at 37 °C, followed by washing and the addition of 100 µL from the tetramethylbenzidine (TMB) substrate. The reaction stopped by the addition of 50 µL of 4N H₂SO₄. Finally, the absorbance was read spectrophotometrically using the Elx 800 bioelisa Reader (Biotek, Winooski VT, USA) at 450 nm. The cut-off was calculated based on the results of known negative samples tested in triplicate (mean OD of negative samples + 3 × SD) and was found to be 0.217. The serum samples of participants were tested in duplicate. Samples with borderline index values (OD/cut-off = 0.8–1.2) were retested in duplicate. In case a discrepancy was found, the results of the three matching ODs were taken as the final result.

### 2.3. Statistical Analysis

Data were coded, entered, and analyzed using IBM SPSS statistics version 23.0 for Windows (SPSS Inc., Chicago, IL, USA). HEV seroprevalence was calculated as the proportion of participants with positive IgG results among all participants with valid blood test results. Results were presented as frequency and percentage of positive HEV serology. The Chi-square test was used to identify the association between HEV seroprevalence and other factors. A *p* value <0.05 was considered statistically significant. To adjust for confounding factors, multiple logistic regression was used. All significant variables in chi-square were included in the regression model.

The Hosmer–Lemeshow goodness-of-fit test was used to check model fitness at *p* < 0.05. Collinearity between independent variables was checked using Spearman correlation. The variables were simultaneously entered into the model. The association was presented as Odds Ratio (OR) together with its 95% Confidence Interval (95%CI).

## 3. Results

### 3.1. Participants Description

We recruited 1329 subjects from the city of Makkah. Participants were from different districts in the city center, the east, the west, the north and the south. Participants’ ages ranged from 8 to 88 years with a median age of 28 and a mean of 30.17 years; the male-to-female ratio was 1.15 (Table 1).

This study cohort included 446/1329 (33.6%) students, 371/1329, (27.9%) janitors and construction workers, housewives (314, 23.6%), and the remaining were clerks, engineers, housekeepers, gatekeepers, and others (198, 14.9%). The housing of the recruited participants included traditional houses (folk houses) (560, 42.1%), apartments (533, 40.1%), shared housing where more than one family shared the same house (132, 9.9%); other housing types included 7.9% of the participants. The participants reported water connection availability in 474 of the houses and its lack of availability in 852 of the houses.

### 3.2. Seroprevalence of HEV

The seroprevalence of HEV IgG in the study participants was 23.8% (316/1329) (Table 2). Males had a higher prevalence than females (29.44 vs. 17.29%, respectively; *p* < 0.001). Age was also significantly associated with increased seroprevalence, showing 32.81%, 17.57%, and 10.00% prevalence in the age groups >30, 11–30, and 0–10 years, respectively (*p* < 0.001). The participants’ occupations were found to be significantly correlated with the seroprevalence of HEV (*p* < 0.001), with the highest prevalence found in housekeepers (42.31%), janitors/construction workers (37.74%) and other occupations collectively (34.78%). The seroprevalence also showed a significant correlation (*p* < 0.001) with the municipalities where the participants lived, with the highest prevalence in the municipalities of the south (39.17%) and east (24.12%). Housing type and the number of individuals living in the same house were also correlated with the HEV seroprevalence (*p* < 0.001) with the highest prevalence in participants living in shared housing (more than one family in the same house), villas, villas with gardens, and others (48.48, 36.0, 35.29 and 29.03%, respectively) and the lowest being in participants living in apartments or traditional houses (22.70% and 17.50%, respectively). Unavailability and/or irregularity of water supply was found to be positively correlated to the seroprevalence of HEV with *p* values of 0.032 and 0.002, respectively.

Using logistic regression, age, gender, and municipality were significantly associated with HEV infection (Table 3); all other variables were not significant. Analysis shows that males were more exposed to HEV infection than females with an odds ratio of 2 (95% CI 1.5–2.6, *p* < 0.001). Logistic regression also showed that individuals in the age group 11–30 years did not show a statistically significant increase in the risk of acquiring the infection than the 0–10 years group (OR 1.9, 95% CI 0.86–4.3, *p* < 0.11), while those in the age group over 30 years were at a 4.5 times higher risk of acquiring the infection than those in the age group 0–10 years of age (95% CI 2–10, *p* < 0.001). Living in certain municipalities in the city—Ajiad (city center), Alsharaei (east), Alshowqia (south) and Alotaibia (city center)— was also associated with a higher chance of acquiring HEV infection than in Alaziziya (east) (OR 8, 95% CI 1.7–37.7, *p* = 0.01; OR 4.3, 95% CI 1.3–14.2, *p* = 0.02; OR 8.6, 95% CI 2.6–28.6, *p* < 0.001; OR 3.5, 95% CI 1.04–11.9, *p* = 0.04; respectively) as shown in Table 3.

## 4. Discussion

Our study is the first community-based HEV seroprevalence study from Saudi Arabia. There are several important and interesting findings from the study. First, it shows that HEV is most likely endemic in Makkah, a large city where millions of pilgrims visit throughout the year from over 182 countries for the Hajj and Umrah pilgrimages. Second, HEV affects all age groups and people from all occupations and socioeconomic backgrounds. Third, HEV significantly affects more males than females. Fourth, those people living in crowded accommodations without a regular supply of water have higher HEV seroprevalence rates.

Our study showed an overall prevalence of 23.8% in the population studied in Makkah. There is a lack of data on the seroprevalence of HEV in Saudi Arabia; thus, comparisons cannot be made with other regions. Available data on HEV from Saudi Arabia are either limited to case studies with small data sets or specific clinical groups such as blood donors or hemodialysis patients [20,21,22,24,25,32,33]. In 1994, a seroepidemiological study of HEV in Riyadh and Gizan of Saudi Arabia showed a seroprevalence of 8.4% in Riyadh and 14.9% in Gizan, with increased prevalence with increased age of participants. The study hypothesized that this high seroprevalence in Gizan was probably due to hygienic conditions and improved sanitation in Riyadh [24]. The study utilized a commercial assay for the detection of HEV-IgG antibodies. Other studies from the same time period for seroprevalence in blood donors showed a prevalence of 16.9% in Jeddah [20]. Studies on acute hepatitis cases showed a seroprevalence of HEV-IgG antibodies of 54.5% in adults in Riyadh [33]. In another study on acute hepatitis in Gizan, the HEV-IgG seroprevalence was found to be 13.7% in children, 16.7% in adolescents, and 11.0% in adults [34].

Recent studies evaluating the prevalence of HEV in blood donors included 806 participants in the Eastern Province of Saudi Arabia [35]. One thousand seventy-eight blood donors in the Qassim region of Saudi Arabia [23] showed a prevalence rate of 3.2% in the Eastern province and 5.7% in Qassim region. Several possible explanations are suggested for the discordance in results between this study and previous reports on HEV prevalence in Saudi Arabia. First, this difference could be due to the difference in study populations. While this study was a population-based study, previous large sample-sized reports were based on recruiting blood donors who were younger in age and healthier individuals. Another potential explanation is the difference in geographical locations, as Saudi Arabia with its large population of more than 33 million inhabitants and vast area of more than two million square kilometers accommodates a variety of cultural and environmental differences and nationalities. While this study was performed in Makkah in the western region where millions of pilgrims gather annually to perform Hajj and Umrah, previous studies were performed in the central province of Qassim and the Eastern province, where there was no large influx of people into these regions from abroad. Finally, the different assays used in the two studies could partially account for such a difference in prevalence. The seroprevalence rate of HEV appears different from that of other Middle East regions where HEV prevalence rates in blood donors are higher: in Yemen, 10.7% [36], Qatar 20.7% [37], Egypt 67.7% [38], Jordan 30.9% [39], and Iran 9.7% [40]. Our results are comparable to the prevalence rate reported from some European countries where the main route of transmission is through consuming pork products. A study in the Netherlands by Verhoef et al., where the sample size was 2494, found a 28.7% prevalence [41]. Teixeira et al. also reported a relatively high prevalence rate in Portugal from a lower sample size (19.9% from 804 participants) [7]. Other studies from Germany (two studies), Spain, the USA (two studies), and the Czech Republic showed a prevalence rate ranging from 2.17% to 16.8% [42,43,44,45,46]. Collectively, these studies demonstrate how the seroprevalence rate of HEV may substantially vary from one country to another.

Results from this study showed an increase in prevalence with age, with the highest prevalence in participants older than 30 years of age with an odds ratio of (*p* < 0.001), which might be due to the longer exposures to the infection risk factors with age. Results also showed a significantly higher prevalence in males than females with an odds ratio of two for being a male (29.44 vs. 17.29%, *p* < 0.001). Older age and male gender were reported in several other studies as associated risk factors for the high prevalence of HEV in different populations [38,46,47]. The living conditions of the recruited subjects played an important role in the spread of the infection, as shown by the higher seroprevalence of HEV in participants living in shared housing and villas (with or without gardens) (*p* < 0.001), in housing where water supply is not available (*p* = 0.032), and in housing where water supply is irregular (*p* = 0.002). The number of house occupants is significantly correlated with the seroprevalence of HEV (*p* < 0.001) which seemed to increase with the number of individuals living in the same house with up to 20 occupants.

HEV genotypes 1 and 2 are transmitted primarily by the fecal-oral route and is responsible for outbreaks in low and middle-income countries where poor sanitation is a transmission risk factor [42,47]. The virus is endemic in several developing countries and widespread in industrialized countries [1]. While the availability of sewage disposal systems in the house is a controversial risk factor for the high seroprevalence of HEV; while some studies reported it as a risk factor for HEV infection [48,49], other studies showed no association between sewage disposal systems and HEV infection [50,51]. In this study, the availability of public sewage disposal connections to the house was not significantly correlated to the high seroprevalence. This might indicate another environmental exposure that accounts for the high seroprevalence in the studied population. 

Animal reservoirs of HEV infection include rabbits, deer, pigs, and wild boars [52,53,54,55,56]. Zoonotic transmission of HEV to humans is documented to occur through several animal species including pigs, rabbits, wild boars, camels, bats, and goats [8,9,14,18,54,55,57,58]. While dromedary camels were reported to be infected with HEV genotype 7 [14,15], their potential to transmit the infection to humans was only reported in a single case from the United Arab Emirates [10]. Recent HEV seroprevalence and genetic characterization studies in dromedary camels from Saudi Arabia [15,29] show an overall seroprevalence of 23.1% with slightly lower prevalence in imported camels compared to domestic camels (22.4% vs. 25.4%, respectively). These studies highlight the potential role of camels as reservoirs for the zoonotic transmission of HEV. Dromedary camels play an important role in the everyday life of most Saudi citizens. They are raised for economic and recreational reasons, and their meat and/or milk are widely consumed by almost all citizens. Whilst our data is novel and advances the dialogue on HEV in Saudi Arabia, a weakness of our study was the absence of data on exposure to domestic animals, including camels. Since the use of IgG assays documents only past HEV infections, future studies are needed to evaluate the burden of active HEV infections using assays to detect IgM-type antibodies and HEV-RNA, with positive RNA samples subjected to nucleic acid sequencing to identify the circulating HEV genotypes.

## 5. Conclusions

In conclusion, our study results provide novel information on community HEV seroprevalence in Makkah and point toward risk factors that might be associated with HEV infection, such as age, gender, water supply, and living conditions. Further studies are required across all regions of Saudi Arabia to estimate the country’s seroprevalence of past or active infection, using tests for HEV IgG, HEV IgM antibodies and/or HEV RNA and underlying determinants of transmission.

## Figures and Tables

**Table 1 viruses-15-00484-t001:** Sociodemographic and environmental characteristics of study population.

Variable		Frequency	Percent
Municipality	Center	638	48.0
	East	427	32.1
	South	217	16.3
	North	8	0.6
	Missing	39	2.9
Gender	Male	710	53.4
	Female	619	46.6
Age Group	0–10	70	5.3
	11–30	683	51.4
	>30	576	43.3
Occupation	Student	446	33.6
	Construction Worker	371	27.9
	Housewife	314	23.6
	Clerks	34	2.6
	Housekeeper	26	2.0
	Others	138	10.3
Residence type	Villa With Garden	17	1.3
	Villa	25	1.9
	Apartment	533	40.1
	Traditional House	560	42.1
	Shared Housing	132	9.9
	Others	16	1.2
	Missing	46	3.5
Water supply	Not Available	852	64.1
	Available	474	35.7
	Missing	3	0.2
Water Regularity	Irregular	1154	86.8
	Regular	167	12.6
	Missing	8	0.6
Sewage	Not Available	698	52.5
	Available	610	45.9
	Missing data	21	1.6

**Table 2 viruses-15-00484-t002:** Correlation of HEV seroprevalence with demographic and environmental factors.

Variables		Negative		Positive		*p* Value	95% CI
		Count	%	Count	%		
Gender	Male	501	70.56%	209	29.44%	<0.001	1.265–1.762
	Female	512	82.71%	107	17.29%		
Age Group	0–10	63	90.00%	7	10.00%	<0.001	–
	11–30	563	82.43%	120	17.57%		
	>30	387	67.19%	189	32.81%		
Occupation	Housekeeper	15	57.69%	11	42.31%	<0.001	–
	Janitor/construction worker	231	62.26%	140	37.74%		
	Clerk	27	79.41%	7	20.59%		
	Housewife	250	79.62%	64	20.38%		
	Student	400	89.69%	46	10.31%		
	Others	90	65.22%	48	34.78%		
Housing Type	Shared Housing	68	51.52%	64	48.48%	<0.001	–
	Villa	16	64.00%	9	36.00%		
	Villa With Garden	11	64.71%	6	35.29%		
	Others	44	70.97%	18	29.03%		
	Apartment	412	77.30%	121	22.70%		
	Traditional House	462	82.50%	98	17.50%		
Number of House Occupants	up to 5	305	78.81%	82	21.19%	<0.001	–
	6–10	469	79.36%	122	20.64%		
	11–20	195	71.69%	77	28.31%		
	>20	44	55.70%	35	44.30%		
Water Availability	Not Available	665	78.05%	187	21.95%	0.032	1.026–1.723
	Available	345	72.78%	129	27.22%		
Water Regularity	Irregular	895	77.56%	259	22.44%	0.002	1.229–2.473
	Regular	111	66.47%	56	33.53%		
Sewage	Not Available	545	78.08%	153	21.92%	0.104	0.965–1.607
	Available	452	74.10%	158	25.90%		
	Missing	16	76.19%	5	23.81%		
Geographical Location	South	132	60.83%	85	39.17%	<0.001	–
	East	324	75.88%	103	24.12%		
	Center	521	81.66%	117	18.34%		

**Table 3 viruses-15-00484-t003:** Binary logistic regression for factors associated with hepatitis E infection.

Variable	OR * (95% CI^+^)	*p* Value
Gender		
Female	Reference	
Male	2 (1.5–2.6)	<0.001
Age group (years)		
0–10	Reference	
11–30	1.9 (0.86–4.3)	0.11
>30	4.5 (2–10)	<0.001
Municipality		
Alaziziya (East)	Reference	
Ajiad (Center)	8 (1.7–37.7)	0.01
Alsharaei (East)	4.3 (1.3–14.2)	0.02
Alshowqia (South)	8.6 (2.6–28.60	<0.001
Alotaibia (Center)	3.5 (1.04–11.9)	0.04

* Odds Ratio, + Confidence Interval.

## Data Availability

All data related to this article are available within the manuscript.

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
