# Peer review of "Hepatitis E Virus (HEV) in Makkah, Saudi Arabia: A Population-Based Seroprevalence Study"

_viruses, 2023, doi:10.3390/v15020484_

Round 1

Reviewer 1 Report

In the present work, Mai M. El-Daly et al. reports findings of HEV seroprevalence in residents of Makkah, Saudi Arabia. It is a first community-based study of its kind in Saudi Arabia, shedding light on differences in HEV seroprevalence among age, sex, living condition, geographic location, etc. groups. The present study employs a commonly used method of ELISA to generate seroprevalence data. Although the findings are interesting and presented in a comprehensive manner, I have some suggestions, questions and concerns that need to be addressed:

First of all, although the article is presented in an intelligible way and is written in standard English, it should be review by a native speaker.

Although data presentation in a table format is a viable option, I think the paper would greatly benefit from presentation of Table 2 data in graphic figures rather than in a single table.

Specific questions and concerns include:

1. In the description of ELISA method (Section 2.2) the authors mention that HEV virus-like particles were used for well coating. However, no information could be found about the origin of these VLPs both in the text or the cited papers (reference no. 16-18). Which HEV genotype was used as the basis for VLP generation? All 3 cited sources investigated HEV seroprevalence in dromedary species. Were these particular VLPs previously used for HEV antibody detection in human species?

2. In Table 3, Age Group category only includes two groups – the reference 0-10 year group and >30 year group. However, a 11-30 year group is missing. Why was this group excluded from regression analysis? Although an increase of HEV seroprevalence in older groups is demonstrated in Table 2, a paper would benefit from the risk assessment of more exclusive age groupings, or, alternatively, an establishment of a risk value associated with increase/decrease in age.

3. Both Tables 2 and 3 present data with respect to geographic location. However, they are titled differently. Do municipalities selected in Table 3 correspond to geographic locations in Table 2 and five clusters where sampling has been carried out as stated in section 2.1.2?

4. Tables 2 and 3 have 3 variables under ‘Sewage’ category: Available, Not Available and Missing. In what way do variables ‘Not Available’ and ‘Missing’ differ? Why are they presented separately?

5. What could be the reason for a significantly higher HEV prevalence rate detected in Makkah, compared to prevalence rates detected in other studies from Saudi Arabia, as stated in Lines 167-170?

Although the seroprevalence findings in Makkah are interesting and comparable with data from both other Middle East regions and countries where HEV is endemic, HEV viral RNA detection in anti-HEV antibody-positive samples, followed by sequencing and phylogenetic analysis would greatly improve the paper and supplement current findings, especially in the light of dromedary-associated HEV-7, prevalent in Saudi Arabia. It would be interesting to know which HEV genotypes dominate in the human population of Saudi Arabia.

Author Response

Reviewer 1

In the present work, Mai M. El-Daly et al. reports findings of HEV seroprevalence in residents of Makkah, Saudi Arabia. It is a first community-based study of its kind in Saudi Arabia, shedding light on differences in HEV seroprevalence among age, sex, living condition, geographic location, etc. groups. The present study employs a commonly used method of ELISA to generate seroprevalence data. Although the findings are interesting and presented in a comprehensive manner, I have some suggestions, questions and concerns that need to be addressed:

  1. First of all, although the article is presented in an intelligible way and is written in standard English, it should be review by a native speaker.

Response: We thank the reviewer for this comment, The manuscript has been reviewed by a native English speaker.

  1. Although data presentation in a table format is a viable option, I think the paper would greatly benefit from presentation of Table 2 data in graphic figures rather than in a single table.

Response: We agree with the reviewer’s comment, but we found that the data is too much to be presented in a figure.

Specific questions and concerns include:

  1. In the description of ELISA method (Section 2.2) the authors mention that HEV virus-like particles were used for well coating. However, no information could be found about the origin of these VLPs both in the text or the cited papers (reference no. 16-18). Which HEV genotype was used as the basis for VLP generation? All 3 cited sources investigated HEV seroprevalence in dromedary species. Were these particular VLPs previously used for HEV antibody detection in human species?

Response: We thank the reviewer for this clarification, we have added the following statement to the methods section “The VLPs (DcHEV-LPs) used for the detection of anti-HEV IgG antibody expressed the partial ORF2 of DcHEV (G7 HEV) by a recombinant baculoviruses expression system. We confirmed that the human anti-G1, G3 and G4 HEV IgG reacted to DcHEV-LPs and each homologous HEV-LPs with similar titers [28]”.

  1. In Table 3, Age Group category only includes two groups – the reference 0-10 year group and >30 year group. However, a 11-30 year group is missing. Why was this group excluded from regression analysis? Although an increase of HEV seroprevalence in older groups is demonstrated in Table 2, a paper would benefit from the risk assessment of more exclusive age groupings, or, alternatively, an establishment of a risk value associated withincrease/decrease in age.

Response: We thank the reviewer for this comment, the age group 11-30 was added to table 3. Our data also show a strong correlation (p<0.001) between seroprevalence and age when divided into 10 years categories.

  1. Both Tables 2 and 3 present data with respect to geographic location. However, they are titled differently. Do municipalities selected in Table 3 correspond to geographic locations in Table 2 and five clusters where sampling has been carried out as stated in section 2.1.2?

Response: We thank the reviewer for this comment, table 3 was modified to reflect the geographical locations of the municipalities included in the study.

  1. Tables 2 and 3 have 3 variables under ‘Sewage’ category: Available, Not Available and Missing. In what way do variables ‘Not Available’ and ‘Missing’ differ? Why are they presented separately?

Response: We thank the reviewer for this comment and would like to bring to the attention of the reviewer that sewage is mentioned only in table 2. The term “Not Available” refers to the unavailability of a public sewage disposal system in the house of residence, while the term “missing” refers to the data being missing upon data entry and has been corrected to “Missing data”.

  1. What could be the reason for a significantly higher HEV prevalence rate detected in Makkah, compared to prevalence rates detected in other studies from Saudi Arabia, as stated in Lines 167-170?

Response: We thank the reviewer for this comment, we have added the potential causes for the difference in seroprevalence of HEV between previous studies and our study including the different assays used as follows “Several possible explanations are suggested for the discordance in results between this study and previous reports on HEV prevalence in Saudi Arabia. First, this difference could be due to the difference in study populations, while this study is a population-based study, previous large sample-sized reports were based on recruiting blood donors who are younger in age and more healthy individuals. Another potential explanation is the difference in geographical locations between this study performed in Makkah in the Western Region where millions of pilgrims gather annually to perform Hajj and Umra and the previous studies performed in the central province of Qassim and the Eastern province, where there is no large flux of people entering these regions from abroad. Finally, the different assays used in the two studies could partially account for such a difference in prevalence.”

  1. Although the seroprevalence findings in Makkah are interesting and comparable with data from both other Middle East regions and countries where HEV is endemic, HEV viral RNA detection in anti-HEV antibody-positive samples, followed by sequencing and phylogenetic analysis would greatly improve the paper and supplement current findings, especially in the light of dromedary-associated HEV-7, prevalent in Saudi Arabia. It would be interesting to know which HEV genotypes dominate in the human population of Saudi Arabia.

Response: We agree with the reviewer’s comment; in fact, the lack of such data about HEV genotypes in the human population in Saudi Arabia has encouraged us to prepare a new prospective study that will address the suggested investigation and will start very soon and the results will be published to fill the gap in knowledge in this area.

Reviewer 2 Report

El-Daly et al., in the article “Hepatitis E Virus (HEV) in Makkah, Saudi Arabia: A Population-Based Seroprevalence Study” performed a community-based serological investigation, using an in-house ELISA, to investigate the HEV seroprevalence in the Makkah population. Anti-HEV IgG were detected with an overall prevalence of 23.8%. A statistical analysis revealed significant correlation between the presence of specific antibodied against HEV and age, occupation, water availability and housing conditions of the subject enrolled in the study.

Although the study provided new information on the seroprevalence of HEV in Makkah, Saudi Arabia and assessed some risk factor associated with anti-HEV IgG seropositivity, there are several points that require attention prior to its publication. Overall, introduction and discussion sections should be improved and statical results of the data need to better exposed. In addition, the English should be refined in several point, all over the manuscript.

All over the manuscript please delete the dot before the references bracket.

In the Introduction (and in Discussion, too) section, the Authors should clearly indicate what HEV genotypes They are referring to.

Line 1: Please add (HEV)

Line 60: Please add appropriate references. In addition, what genotypes are circulating in this geographic area?

In the Materials and Methods section, please reword the introduction of the 2.1. Study Design section and the 2.1.2. Population and sampling site section.

Did the Authors compare the performance of the in-house ELISA test used in this study to the most common commercial serological kits employed to detected anti-HEV antibodies?

Regarding the Results, please move the 3.1. Participants Description section to Materials and Methods. In addition, the statistical data need a revision. In the 3.2. Seroprevalence of HEV section and in the table 2 when the Authors referred to p-value, for each group of factor risk, it is necessary to establish a reference to compare the other condition assessed.

In Table 3, some municipalities were introduced for the first time. Are they located in center, east, south or north?

Did the Authors have a hypothesis about the difference on prevalence obtained, compared to the other study performed in Saudi Arabia [12,21]? How the use of a commercial kit vs an in-house ELISA can affect the result of the investigation?

Author Response

Reviewer 2

El-Daly et al., in the article “Hepatitis E Virus (HEV) in Makkah, Saudi Arabia: A Population-Based Seroprevalence Study” performed a community-based serological investigation, using an in-house ELISA, to investigate the HEV seroprevalence in the Makkah population. Anti-HEV IgG were detected with an overall prevalence of 23.8%. A statistical analysis revealed significant correlation between the presence of specific antibodies against HEV and age, occupation, water availability and housing conditions of the subject enrolled in the study.

  1. Although the study provided new information on the seroprevalence of HEV in Makkah, Saudi Arabia and assessed some risk factor associated with anti-HEV IgG seropositivity, there are several points that require attention prior to its publication. Overall, introduction and discussion sections should be improved and statical results of the data need to better exposed. In addition, the English should be refined in several point, all over the manuscript.

Response: We thank the reviewer for this comment, we have modified the introduction methods and discussion sections according to the specific comments of the reviewers and revised the English language of the manuscript.

  1. All over the manuscript please delete the dot before the references bracket.

Response: We apologize for this misformatting of the citations, the reference citations have been reviewed and are now in the correct format.

  1. In the Introduction (and in Discussion, too) section, the Authors should clearly indicate what HEV genotypes They are referring to.

Response: We thank the reviewer for this comment, despite the few reports in the Middle East on the HEV genotypes we have added a statement about the genotypes in the introduction section “Although several countries in The Middle East have recorded medium to high HEV seroprevalence rates, reports on HEV genotypes are rare. HEV genotype 7 was first reported in dromedary camels by Woo et al [12] in the UAE from feces samples. Later in 2016, chronic zoonotic transmission with HEV genotype 7 was reported in a liver transplant patient who regularly consumed camel meat and milk in the UAE [8]. We recently reported the genetic diversity of HEV genotype 7 in imported and domestic dromedary camels in Saudi Arabia [13]. Our results showed that the sequences generated from African domestic dromedary camel samples clustered with Genbank sequences from Kenya, Somalia, and UAE; while sequences generated from domestic dromedary camels clustered with Genbank sequences from UAE, and both domestic and imported sequences clustered away from isolates reported from Pakistan. Reports from Tunisia and Israel showed the detection of HEV genotypes 1 and 3 in wastewater samples [14, 15] and reports from Israel showed HEV genotype 1 in acute HEV cases [16].”.

  1. Line 1: Please add (HEV)

Response: We thank the reviewer for this comment and would like to bring attention to the insertion of (HEV) in the title of the manuscript.

  1. Line 60: Please add appropriate references. In addition, what genotypes are circulating in this geographic area?

Response: We thank the reviewer for this comment; the references were added and the few reports in the Middle East on HEV genotypes have been added.

  1. In the Materials and Methods section, please reword the introduction of the 2.1. Study Design section and the 2.1.2. Population and sampling site section.

Response: We apologize for this typing error, the introductions of section 2.1 was rephrased to “This cross-sectional community-based household seroprevalence study was conducted in the city of Makkah, Saudi Arabia.” and section 2.1.2 to “The study included residents of Makkah city and a multi-stage stratified cluster sampling was used to include the participating households”.

  1. Did the Authors compare the performance of the in-house ELISA test used in this study to the most common commercial serological kits employed to detected anti-HEV antibodies?

Response: We thank the reviewer for this comment, the in-house assay was used before for seroprevalence studies in dromedary camels and used G3 HEV-LPs as an antigen that previously showed a higher sensitivity for HEV IgM, IgA, and IgG than the commercial kits. The following statement has been added to the methods section “The in-house ELISA use G3 HEV-LPs as antigen previously showed a higher sensitivity for HEV IgM, IgA, and IgG than the commercial kits [29].”.

  1. Regarding the Results, please move the 3.1. Participants Description section to Materials and Methods. In addition, the statistical data need a revision. In the 3.2. Seroprevalence of HEV section and in the table 2 when the Authors referred to p-value, for each group of factor risk, it is necessary to establish a reference to compare the other condition assessed.

Response: We thank the reviewer for this comment, The participant descriptions were added to the methods section as follows “Subjects were recruited 1329 subjects from the different districts in the city center, the east, the west, the north and the south of the city of Makkah.”.

In the methods section, we have more detail to statistical methods to describe the details of the multiple logistic regression model as follows “Hosmer–Lemeshow goodness-of-fit test was used to check model fitness at p < 0.05. col-linearity between independent variables was checked using spearman correlation. The variables were simultaneously entered into the model. The association was presented as Odds Ratio (OR) together with its 95% Confidence Interval (CI).”

  1. In Table 3, some municipalities were introduced for the first time. Are they located in center, east, south or north?

Response: We apologize for this confusion, table 3 was modified to reflect the geographical locations mentioned in table 2.

  1. Did the Authors have a hypothesis about the difference on prevalence obtained, compared to the other study performed in Saudi Arabia [12,21]? How the use of a commercial kit vs an in-house ELISA can affect the result of the investigation?

Response: We thank the reviewer for this comment, we have added the potential causes for the difference in seroprevalence of HEV between previous studies and our study including the different assays used as follows “Several possible explanations are suggested for the discordance in results between this study and previous reports on HEV prevalence in Saudi Arabia. First, this difference could be due to the difference in study populations, while this study is a population-based study, previous large sample-sized reports were based on recruiting blood donors who are younger in age and more healthy individuals. Another potential explanation is the difference in geographical locations as Saudi Arabia with its large population of more than 33 million inhabitants and vast area of more than 2 million square kilometers accommodates a variety of cultural and environmental differences and nationalities. While this study was performed in Makkah in the Western Region where millions of pilgrims gather annually to perform Hajj and Umra, previous studies were performed in the central province of Qassim and the Eastern province, where there is no large flux of people entering these regions from abroad. Finally, the different assays used in the two studies could partially account for such a difference in prevalence.”  

Reviewer 3 Report

The authors investigated community-based cross-sectional HEV seroprevalence in Makkah, Saudi Arabia. Anti-HEV IgG antibodies were detected in sera using ELISA. The frequency of HEV serology and its correlation with demographic, and environmental factors were evaluated. 1329 individuals were enrolled. Males had significantly higher seroprevalence than females. Seroprevalence had significant correlations with age, occupation, lack of regular water supply and housing conditions. The authors’ results show that HEV is endemic in Makkah and affects all age groups and occupations. HEV affects more males than females and those living in crowded accommodations without a regular supply of water. The manuscript was generally well written. These information is worth being published in Viruses. It is better, if the authors add some discussion.

Accuracy or sensitivity of various HEV IgG ELISA kits are different. In this study, all sera are measured by the same method, and comparison of seropositivity between different populations is valid. However, difference of accuracy or sensitivity of various ELISA kits should be argued when the prevalence in this study is compared with those in other studies. The authors should discuss it in the second paragraph of Discussion section.

Author Response

Reviewer 3

  1. The authors investigated community-based cross-sectional HEV seroprevalence in Makkah, Saudi Arabia. Anti-HEV IgG antibodies were detected in sera using ELISA. The frequency of HEV serology and its correlation with demographic, and environmental factors were evaluated. 1329 individuals were enrolled. Males had significantly higher seroprevalence than females. Seroprevalence had significant correlations with age, occupation, lack of regular water supply and housing conditions. The authors’ results show that HEV is endemic in Makkah and affects all age groups and occupations. HEV affects more males than females and those living in crowded accommodations without a regular supply of water. The manuscript was generally well written. These information is worth being published in Viruses. It is better, if the authors add some discussion.

Response: We thank the reviewer for the positive feedback.

  1. Accuracy or sensitivity of various HEV IgG ELISA kits are different. In this study, all sera are measured by the same method, and comparison of seropositivity between different populations is valid. However, difference of accuracy or sensitivity of various ELISA kits should be argued when the prevalence in this study is compared with those in other studies. The authors should discuss it in the second paragraph of Discussion section.

Response: We thank the reviewer for this comment, we have added the potential causes for the difference in seroprevalence of HEV between previous studies and our study including the different assays used as follows “Several possible explanations are suggested for the discordance in results between this study and previous reports on HEV prevalence in Saudi Arabia. First, this difference could be due to the difference in study populations, while this study is a population-based study, previous large sample-sized reports were based on recruiting blood donors who are younger in age and more healthy individuals. Another potential explanation is the difference in geographical locations as Saudi Arabia with its large population of more than 33 million inhabitants and vast area of more than 2 million square kilometers accommodates a variety of cultural and environmental differences and nationalities. While this study was performed in Makkah in the Western Region where millions of pilgrims gather annually to perform Hajj and Umra, previous studies were performed in the central province of Qassim and the Eastern province, where there is no large flux of people entering these regions from abroad. Finally, the different assays used in the two studies could partially account for such a difference in prevalence.”

Reviewer 4 Report

Dear Authors,

your paper reports the serological prevalence of HEV in human population from a large city in Saudi Arabia and analyse some risk factors.

The results could fill a gap in knowledge of the epidemiology of HEV in Middle Eastern countries.

However, to be published, the manuscript requires significant modifications (particularly in the statistical analysis).

For these reasons, I suggest the Authors to take into account the following comments.

Major concerns

Introduction. The reader unfamiliar with the subject should find some indication of the virus and its epidemiology ( please pay attention to the recent change in classification). In particular, the epidemiology of the different genotypes of the virus in developed and undeveloped countries, the fact that only certain genotypes are zoonotic, the mode of transmission of the different genotypes, etc. should be reported.

If available, information on HEV genotypes present in Saudi Arabia should be provided.

 L. 47. Specify that the infection is frequently asymptomatic.

 L. 54-55. Specify which HEV genotypes are transmitted via contaminated water and which genotypes via contaminated meat or other animal products (specifying the origin of the food).

 L. 57. Unfortunately, “care” is not enough. The products must be eaten cooked.

 L. 63. Specify which genotypes are endemic in Saudi Arabia.

 L. 64-65. Specify better the aims of the work (e.g. risk factor evaluation).

 L. 68. Provide some information on the city of Makkah (e.g. number of inhabitants, geographical location, etc.). Suggest moving here the text to lines 157-158.

 L. 69. It is unclear what the authors mean by "public" (general uninfected population?).

 L. 80. Based on what factors was Makkah city divided into five clusters? (e.g. size of the cluster, number of inhabitants in the cluster, social characteristics, demographics, etc.). This paragraph should explain what individual information (age, work, etc.) was collected and how (questionnaire or interview?).

 L. 88. Please specify that an in-house ELISA was used.

 L. 90. Specify the HEV genotype used to produce the VLPs.

 L. 104-112. The modalities of the statistical analysis must be modified. I suggest performing (as has been partially done) a two-stage-analysis. In the first stage, the categorical variables are to be screened using the χ2 test. In the second stage, the factors which were screened through (p < 0.10) must be evaluated using unconditional multiple logistic regression. The goodness of fit of the models must be assessed based on the likelihood-ratio statistic and the Hosmer-Lemeshow statistic. The type of model must be specified (e.g. simultaneous entry of all the variables, backward stepwise, etc.). Before the multiple logistic regression, the correlation matrix must be examined for potential collinearity among the independent variables. The software used by the authors is capable of performing these analyses.

 L. 117. Also provide the median age

 Table 1. It is not clear to me why the number of subjects examined according to 4 Municipalities is given in the table. The authors had previously written that the clusters used for sampling were 5. What am I missing? Why were only 8 samples collected in Municipality North?

On what basis were the age classes constructed?

 L. 122. Briefly explain what is meant by traditional houses, apartments, shared housing, villa, etc. (see also my comment at L. 80).

Table 2. In this table, there are some variables not present in Table 1 and not described earlier in the text (e.g. why did the authors investigate the variable "pest control"? Do they consider it a risk factor? If so, on what basis? The same considerations also apply to the variable "Diabetes" with only 8 subjects). I also suggest indicating the confidence intervals.

Table 3. I suggest using all statistically associated variables in the logistic regression (see also my comment on L. 104-112).Why were only 2 age classes used?

Discussion. This paragraph should be rewritten emphasising the main aspects of the results obtained. The comparison with seroprevalence data from European countries must be made considering that in these countries most infections are caused by the zoonotic genotypes HEV-3 and HEV-4 transmitted mainly through the consumption of pork.

The age-related increase in prevalence is an expected result, but the causes must be reported.

L.200-202. This result is interesting. Do the authors think it is possible to assume the presence of an animal reservoir of HEV in Saudi Arabia? If so, which one? Are there any data on HEV infection in animals in Saudi Arabia?

Minor concerns

L. 46. Suggest adding "(HEV)" after "Hepatitis E Virus".

L. 50 and throughout the text. Insert a full stop "." after the bibliography. E.g. '...pregnancy[2, 3].' and not 'pregnancy.[2, 3]'.

L. 104. Replace 'Statistical methods' with 'Statistical analysis'.

L. 110. Replace "p" with "p" (in italics).

Author Response

Reviewer 4

Dear Authors,

your paper reports the serological prevalence of HEV in human population from a large city in Saudi Arabia and analyse some risk factors.

The results could fill a gap in knowledge of the epidemiology of HEV in Middle Eastern countries.

However, to be published, the manuscript requires significant modifications (particularly in the statistical analysis).

For these reasons, I suggest the Authors to take into account the following comments.

Major concerns

  1. The reader unfamiliar with the subject should find some indication of the virus and its epidemiology ( please pay attention to the recent change in classification). In particular, the epidemiology of the different genotypes of the virus in developed and undeveloped countries, the fact that only certain genotypes are zoonotic, the mode of transmission of the different genotypes, etc. should be reported.

Response: We thank the reviewer for this clarification, the following paragraph was added to the introduction section “Although several countries in The Middle East have recorded medium to high HEV seroprevalence rates, reports on HEV genotypes are rare. HEV genotype 7 was first reported in dromedary camels by Woo et al [12] in the UAE from feces samples. Later in 2016, chronic zoonotic transmission with HEV genotype 7 was reported in a liver transplant patient who regularly consumed camel meat and milk in the UAE [8]. We recently reported the genetic diversity of HEV genotype 7 in imported and domestic dromedary camels in Saudi Arabia [13]. Our results showed that the sequences generated from African domestic dromedary camel samples clustered with Genbank sequences from Kenya, Somalia, and UAE; while sequences generated from domestic dromedary camels clustered with Genbank sequences from UAE, and both domestic and imported sequences clustered away from isolates reported from Pakistan. Reports from Tunisia and Israel showed the detection of HEV genotypes 1 and 3 in wastewater samples [14, 15] and reports from Israel showed HEV genotype 1 in acute HEV cases [16]. In Saudi Arabia, there is scanty information on the prevalence of HEV in the general population [17]. Reports utilizing small sample sizes and targeting selected populations such as blood donors and hemodialysis patients [18-23] indicate that HEV is endemic in Saudi Arabia.”.

  1. If available, information on HEV genotypes present in Saudi Arabia should be provided.

Response: We agree with the reviewer on the significance of this information, but we found no data on the HEV genotypes in Saudi Arabia except our previous study on HEV genetic diversity in dromedary camels. We have added it in the introduction section together with other studies reporting the HEV genotypes in the Middle East as follows “Although several countries in The Middle East have recorded medium to high seroprevalence rates, the reports on HEV genotypes are rare. Reports from Tunisia and Israel showed the detection of HEV genotypes 1 and 3 in wastewater samples [9, 10] and reports from Israel showed HEV Genotype 1 I acute HEV cases [11]. Only one report of Chronic HEV genotype 7 zoonotic transmission was reported in the UAE [5]. HEV genotype 7 was reported in dromedaries in Saudi Arabia [12] and the UAE [13].”.

  1. 47. Specify that the infection is frequently asymptomatic.

Response: We thank the reviewer for this comment, the statement has been changed to “The infection is mostly asymptomatic and self-limited in the population”.

  1. 54-55. Specify which HEV genotypes are transmitted via contaminated water and which genotypes via contaminated meat or other animal products (specifying the origin of the food).

Response: We thank the reviewer for this comment, we have modified the statement to include HEV genotypes as follows “HEV is mainly transmitted through the fecal-oral route by drinking contaminated water in endemic areas where HEV genotypes 1 and 2 are most prevalent [5], or by consuming contaminated animal products, particularly pork, in industrialized countries where genotypes 3 and 4 are most prevalent [6, 7].”.

  1. 57. Unfortunately, “care” is not enough. The products must be eaten cooked.

Response: We thank the reviewer for this comment, the statement has been changed to “and cooking the consumed animal products [10]”.

  1. 63. Specify which genotypes are endemic in Saudi Arabia.

Response: We thank the reviewer for this comment; unfortunately, there is no data on the genotypes of HEV in Saudi Arabia except our study in dromedary camels, but we have added this statement in the introduction section “Although several countries in The Middle East have recorded medium to high HEV seroprevalence rates, reports on HEV genotypes are rare. HEV genotype 7 was first reported in dromedary camels by Woo et al [12] in the UAE from feces samples. Later in 2016, chronic zoonotic transmission with HEV genotype 7 was reported in a liver transplant patient who regularly consumed camel meat and milk in the UAE [8]. We recently reported the genetic diversity of HEV genotype 7 in imported and domestic dromedary camels in Saudi Arabia [13]. Our results showed that the sequences generated from African domestic dromedary camel samples clustered with Genbank sequences from Kenya, Somalia, and UAE; while sequences generated from domestic dromedary camels clustered with Genbank sequences from UAE, and both domestic and imported sequences clustered away from isolates reported from Pakistan. Reports from Tunisia and Israel showed the detection of HEV genotypes 1 and 3 in wastewater samples [14, 15] and reports from Israel showed HEV genotype 1 in acute HEV cases [16]”.

  1. 64-65. Specify better the aims of the work (e.g. risk factor evaluation).

Response: We thank the reviewer for this comment, the statement has been changed to “In this study, we performed a community-based cross-sectional study to determine the seroprevalence and evaluate potential risk factors of hepatitis E virus in residents of Makkah, Saudi Arabia.”.

  1. 68. Provide some information on the city of Makkah (e.g. number of inhabitants, geographical location, etc.). Suggest moving here the text to lines 157-158.

Response: We thank the reviewer for this comment, we have added the following statement to the introduction section “Makkah is the holiest city in the Muslim World, where millions of Muslims from around the world gather every year to perform the Hajj pilgrimage rituals in a certain time of the year and millions more to perform Umrah all year round. The city has an area of about 1200 km2 and a population of about 1.6 million inhabitants.”.

  1. 69. It is unclear what the authors mean by "public" (general uninfected population?).

Response: We thank the reviewer for this comment, the statement was removed from the text.

  1. 80. Based on what factors was Makkah city divided into five clusters? (e.g. size of the cluster, number of inhabitants in the cluster, social characteristics, demographics, etc.). This paragraph should explain what individual information (age, work, etc.) was collected and how (questionnaire or interview?).

Response: We thank the reviewer for this comment, We have added the following two statements to the methods section “Makkah city was divided into five clusters based on sociodemographic characteristics” and a description of the data collection tool “A semi-structured questionnaire was used to collect sociodemographic data such as age, gender, housing type, etc., and environmental risk factors such as connection to the sewage network, and availability of clean water supply. The questionnaire was administered by one of the investigators.”.

  1. 88. Please specify that an in-house ELISA was used.

Response: We thank the reviewer for this comment, the statement has been changed to “HEV IgG antibodies were detected in the collected samples using an in-house enzyme-linked immunoassay (ELISA)”.

  1. 90. Specify the HEV genotype used to produce the VLPs.

Response: We thank the reviewer for this clarification, the following statement was added to the methods section “The VLPs (DcHEV-LPs) used for the detection of anti-HEV IgG antibody expressed the partial ORF2 of DcHEV (G7 HEV) by a recombinant baculoviruses expression system. We confirmed that the human anti-G1, G3 and G4 HEV IgG reacted to DcHEV-LPs and each homologous HEV-LPs with similar titers [28]. The in-house ELISA uses HEV-LPs as an antigen and has previously shown a higher sensitivity for HEV IgM, IgA, and IgG than the commercial kits [29].”.

  1. 104-112. The modalities of the statistical analysis must be modified. I suggest performing (as has been partially done) a two-stage-analysis. In the first stage, the categorical variables are to be screened using the χ2 test. In the second stage, the factors which were screened through (p < 0.10) must be evaluated using unconditional multiple logistic regression. The goodness of fit of the models must be assessed based on the likelihood-ratio statistic and the Hosmer-Lemeshow statistic. The type of model must be specified (e.g. simultaneous entry of all the variables, backward stepwise, etc.). Before the multiple logistic regression, the correlation matrix must be examined for potential collinearity among the independent variables. The software used by the authors is capable of performing these analyses.

Response: We thank the reviewer for this comment, The goodness of fit of the model, collinearity checking, and the type of model were added in the statistical analysis section

  1. 117. Also provide the median age

Response: We thank the reviewer for this comment, the median age (28 years) has been added.

  1. Table 1. It is not clear to me why the number of subjects examined according to 4 Municipalities is given in the table. The authors had previously written that the clusters used for sampling were 5. What am I missing? Why were only 8 samples collected in Municipality North?

Response: We apologize for this confusion; we have added the geographical location of the municipalities in table 3.

  1. On what basis were the age classes constructed?

Response: We thank the reviewer for this comment, the age was grouped into 3 groups to differentiate children of young age <10 years from young adults 11-30 years, and older adults >30 years.

  1. 122. Briefly explain what is meant by traditional houses, apartments, shared housing, villa, etc. (see also my comment at L. 80).

Response: We thank the reviewer for this comment, we have added the following explanatory comments “traditional houses (Folk houses), and shared housing (more than one family living in the same house)”.

  1. Table 2. In this table, there are some variables not present in Table 1 and not described earlier in the text (e.g. why did the authors investigate the variable "pest control"? Do they consider it a risk factor? If so, on what basis? The same considerations also apply to the variable "Diabetes" with only 8 subjects). I also suggest indicating the confidence intervals.

Response: We thank the reviewer for this comment, the variables pest control and diabetes were removed from table 2

  1. Table 3. I suggest using all statistically associated variables in the logistic regression (see also my comment on L. 104-112). Why were only 2 age classes used?

Response: We thank the reviewer for this comment, Age group 11-30 results were added to the table.

  1. This paragraph should be rewritten emphasising the main aspects of the results obtained. The comparison with seroprevalence data from European countries must be made considering that in these countries most infections are caused by the zoonotic genotypes HEV-3 and HEV-4 transmitted mainly through the consumption of pork.

Response: We thank the reviewer for this comment, the whole paragraph has been rewritten to explain the causes of the high seroprevalence in our study compared to previously reported studies and other factors.

  1. The age-related increase in prevalence is an expected result, but the causes must be reported.

Response: We thank the reviewer for this comment, the statement has been modified to “Results from this study showed an increase in prevalence with age with the highest prevalence in participants older than 30 years of age with an odds ratio of (p<0.001) which might be due to the longer exposures to the infection risk factors with age.”.

  1. 200-202. This result is interesting. Do the authors think it is possible to assume the presence of an animal reservoir of HEV in Saudi Arabia? If so, which one? Are there any data on HEV infection in animals in Saudi Arabia?

Response: We thank the reviewer for this comment, we have previously described the seroprevalence and genetic characterization of HEV in dromedary camels in two of our previous studies. The following statement has been added to the discussion section” Dromedary camels play an important role in the everyday life of most Saudi citizens. They are raised for economical, recreational reasons and their meat and/or milk are widely consumed by almost all citizens.”.

Minor concerns

  1. 46. Suggest adding "(HEV)" after "Hepatitis E Virus".

Response: We thank the reviewer for this comment and would like to bring attention to the insertion of (HEV) in the title of the manuscript.

  1. 50 and throughout the text. Insert a full stop "." after the bibliography. E.g. '...pregnancy[2, 3].' and not 'pregnancy.[2, 3]'.

Response: We apologize for this misformatting of the citations, the reference citations have been reviewed and are now in the correct format.

  1. 104. Replace 'Statistical methods' with 'Statistical analysis'.

Response: The section title has been changed to “Statistical Analysis”.

  1. 110. Replace "p" with "p" (in italics).

Response: "p" has been replaced with "p" (in italics).

Round 2

Reviewer 2 Report

After revisions, I found that the final version of the manuscript is suitable for publication.

Author Response

After revisions, I found that the final version of the manuscript is suitable for publication.

Response: We thank the reviewer for the positive feedback.

Reviewer 4 Report

Dear Authors,
Thank you for taking many of my suggestions into consideration. However, some issues that I had indicated in the previous revision were not adequately answered.

Abstract
L. 29. Plese add "(HEV)" after "Hepatitis E virus".
L. 29. Replace "little" with "Little".
L. 32. Add "an in-house" before ELISA.
L. 34. Add median age value.
L. 36 and elsewhere in the text (e.g. L. 165, 167, 169): Replace "p" with "p" (in italics).

Introduction
As previously indicated, some information concerning HEV (classification, type of nucleic acid, number of genotypes, modes of transmission, animal reservoirs, etc.) should be briefly presented in this paragraph.

L. 46. Plese add "(HEV) "after "Hepatitis E virus".

Materials and Methods
L. 118. Replace "(G7 HEV)" with "(HEV-7)".
L. 119. Replace "... human anti-G1, G3 and G4 HEV IgG" with " ... human IgG anti HEV-1, HEV-3 and HEV-4".

Statistical Analysis
L. 144. Replace "col-linearity" with "Collinearity".
L. 145. Replace "spearman" with "Spearman".
L. 147. Replace '(CI)' with '(95%CI)'.

3.2. Seroprevalence of HEV and Table 2.
In this paragraph and in Table 2 there is a serious methodological error that I had not noticed in the previous revision (and I apologise for this to the Authors). For example, the authors write "Males had a higher prevalence than females (66.1 vs 33.9%, respectively)". However, the reported prevalence values are column percentages and not row percentages (as would be correct). In epidemiology, prevalence is the proportion of a particular population (eg. Male or Female) found to be affected by a disease at a specific time. It is derived by comparing the number of people found to have the condition with the total number of people studied. In this case, the prevalence in males should be calculated as 209/701=29.4% and that of females as 107/619=17.3%.
The same consideration applies to all prevalence values given in this paragraph. The authors must calculate all prevalence values correctly and amend the text and table accordingly. The 95%CI must be reported in the table.
Why were only 8 samples collected in Municipality North?

L.182-189. I believe that it makes little sense to subject to logistic regression analysis only certain variables which, on univariate analysis, are significantly associated with an increase in prevalence (e.g. Number of House Occupants, Water Availability, etc.). These variables must be included in the regression analysis. Alternatively, the authors should explain why only the variables Gender, Age class, and Geographical location were used).

L.242. The OR value is missing.
L. 256. The Authors write "HEV is transmitted primarily by the faecal-oral route". This is only correct for HEV-1 and -2.

Author Response

Dear Authors,

Thank you for taking many of my suggestions into consideration. However, some issues that I had indicated in the previous revision were not adequately answered.

Abstract

  1. 29. Please add "(HEV)" after "Hepatitis E virus".

Response: We thank the reviewer for this comment, (HEV) has been added.

  1. 29. Replace "little" with "Little".

Response: We thank the reviewer for this comment, "little" has been replaced with "Little".

  1. 32. Add "an in-house" before ELISA.

Response: We thank the reviewer for this comment, "an in-house" has been added.

  1. 34. Add median age value.

Response: We thank the reviewer for this comment, median age has been added.

  1. 36 and elsewhere in the text (e.g. L. 165, 167, 169): Replace "p" with "p" (in italics).

Response: We thank the reviewer for this comment, “p” has been replaced with “p” (in italics) all over the manuscript.

Introduction

  1. As previously indicated, some information concerning HEV (classification, type of nucleic acid, number of genotypes, modes of transmission, animal reservoirs, etc.) should be briefly presented in this paragraph.

Response: We thank the reviewer for this comment, the following statement was added to the introduction section “Hepatitis E virus (HEV) is a positive-sense single-stranded RNA virus that belongs to the genus Orthohepevirus in the Hepeviridae family [1]. The viral genome is about 7.2 kB in length and is organized into 3 open reading frames with at least eight genotypes (HEV-1 to HEV-8) [2].”. modes of transmission are reported in lines 60-66 as follows “HEV is mainly transmitted through the fecal-oral route by drinking contaminated water in endemic areas where HEV genotypes 1 and 2 are most prevalent [6], or by consuming contaminated animal products, particularly pork, in industrialized countries where genotypes 3 and 4 are most prevalent [7, 8]. Animal reservoirs for HEV transmission were also reported in wild pigs, mongooses, deer, rabbits, [7, 8] and dromedary camels in the Middle East [9]. Vertical transmission from infected mothers to their infants can occur [10, 11].”

  1. 46. Please add "(HEV) "after "Hepatitis E virus".

Response: We thank the reviewer for this comment, “HEV” has been added.

Materials and Methods

  1. 118. Replace "(G7 HEV)" with "(HEV-7)".

Response: We thank the reviewer for this comment, "(G7 HEV)" has been replaced with "(HEV-7)".

  1. 119. Replace "... human anti-G1, G3 and G4 HEV IgG" with " ... human IgG anti HEV-1, HEV-3 and HEV-4".

Response: We thank the reviewer for this comment, "... human anti-G1, G3 and G4 HEV IgG" has been replaced with " ... human IgG anti HEV-1, HEV-3 and HEV-4".

Statistical Analysis

  1. 144. Replace "col-linearity" with "Collinearity".

Response: We thank the reviewer for this comment, "col-linearity" has been replaced with "Collinearity".

  1. 145. Replace "spearman" with "Spearman".

Response: We thank the reviewer for this comment, "spearman" has been replaced with "Spearman"

  1. 147. Replace '(CI)' with '(95%CI)'.

Response: We thank the reviewer for this comment, '(CI)' has been replaced with '(95%CI)'.

3.2. Seroprevalence of HEV and Table 2.

  1. In this paragraph and in Table 2 there is a serious methodological error that I had not noticed in the previous revision (and I apologise for this to the Authors). For example, the authors write "Males had a higher prevalence than females (66.1 vs 33.9%, respectively)". However, the reported prevalence values are column percentages and not row percentages (as would be correct). In epidemiology, prevalence is the proportion of a particular population (eg. Male or Female) found to be affected by a disease at a specific time. It is derived by comparing the number of people found to have the condition with the total number of people studied. In this case, the prevalence in males should be calculated as 209/701=29.4% and that of females as 107/619=17.3%.

Response: We thank the reviewer for this comment and apologize for this confusion, the table has been modified accordingly and the text as well.

  1. The same consideration applies to all prevalence values given in this paragraph. The authors must calculate all prevalence values correctly and amend the text and table accordingly. The 95%CI must be reported in the table.

Response: We thank the reviewer for this comment and apologize for this confusion, the text was modified according to the corrected values and the 95%CI was added.

  1. Why were only 8 samples collected in Municipality North?

Response: We thank the reviewer for this comment, only 8 participants in this municipality agreed to participate in the study.

  1. 182-189. I believe that it makes little sense to subject to logistic regression analysis only certain variables which, on univariate analysis, are significantly associated with an increase in prevalence (e.g. Number of House Occupants, Water Availability, etc.). These variables must be included in the regression analysis. Alternatively, the authors should explain why only the variables Gender, Age class, and Geographical location were used).

Response: We thank the reviewer for this comment, all the significant variables in the chi-square test were included in the logistic regression analysis. Only the variables that showed significant results in the logistic regression analysis were listed in table 3 and shown in the results section. In the statistical analysis methods, we have added this statement “All significant variables in chi-square were included in the regression model.”. In the results section we have modified the logistic regression statement to “Using logistic regression, age, gender, and municipality were significantly associated with HEV infection (Table 3) all other variables were not significant.”.

  1. 242. The OR value is missing.

Response: We thank the reviewer for this comment, the logistic regression analysis has been modified as follows “Using logistic regression, age, sex, and municipality were significantly associated with HEV infection (Table 3). Analysis shows that males are more exposed to HEV infection than females with an odds ratio of 2 (95% CI 1.5-2.6, p<0.001). Logistic regression also showed that individuals in the age group of 11-30 years did not show a statistically significant increase in the risk of acquiring the infection than the 0-10 years group (OR 1.9, 95% CI 0.86-4.3, p<0.11) while those at the age of more than 30 years are at a 4.5 times higher risk of acquiring the infection than those in the age group 0-10 years of age (95% CI 2-10, p<0.001). Living in certain municipalities in the city Ajiad, Alsharaei, Alshowqia and Alotaibia is also associated with more chance of acquiring HEV infection than Al-aziziya (OR 8, 95% CI 1.7-37.7, p=0.01; OR 4.3, 95% CI 1.3-14.2, p=0.02; OR 8.6, 95% CI 2.6-28.6, p<0.001; OR 3.5, 95% CI 1.04-11.9, p=0.04; respectively) as shown in table 3.”.

  1. 256. The Authors write "HEV is transmitted primarily by the faecal-oral route". This is only correct for HEV-1 and -2.

Response: We apologize for this error; the statement has been changed to “HEV genotypes 1 and 2 are transmitted primarily by the fecal-oral route”.